# Synergistic electroreduction of carbon dioxide to carbon monoxide on bimetallic layered conjugated metal-organic frameworks

Haixia Zhong[1,7], Mahdi Ghorbani-Asl [2,7], Khoa Hoang Ly[1,7], Jichao Zhang[3,7], Jin Ge[2], Mingchao Wang[1], Zhongquan Liao[4], Denys Makarov [2], Ehrenfried Zschech[4], Eike Brunner[1], Inez M. Weidinger[1], Jian Zhang[1,5], Arkady V. Krasheninnikov[2,6], Stefan Kaskel [1], Renhao Dong [1,8] & Xinliang Feng [1,8✉]

Highly effective electrocatalysts promoting $CO_2$ reduction reaction ($CO_2RR$) is extremely desirable to produce value-added chemicals/fuels while addressing current environmental challenges. Herein, we develop a layer-stacked, bimetallic two-dimensional conjugated metal-organic framework (2D c-MOF) with copper-phthalocyanine as ligand ($CuN_4$) and zinc-bis (dihydroxy) complex ($ZnO_4$) as linkage (PcCu-$O_8$-Zn). The PcCu-$O_8$-Zn exhibits high CO selectivity of 88%, turnover frequency of $0.39\,s^{-1}$ and long-term durability (>10 h), surpassing thus by far reported MOF-based electrocatalysts. The molar $H_2$/CO ratio (1:7 to 4:1) can be tuned by varying metal centers and applied potential, making 2D c-MOFs highly relevant for syngas industry applications. The contrast experiments combined with *operando* spectroelectrochemistry and theoretical calculation unveil a synergistic catalytic mechanism; $ZnO_4$ complexes act as $CO_2RR$ catalytic sites while $CuN_4$ centers promote the protonation of adsorbed $CO_2$ during $CO_2RR$. This work offers a strategy on developing bimetallic MOF electrocatalysts for synergistically catalyzing $CO_2RR$ toward syngas synthesis.

[1] Center for Advancing Electronics Dresden (Cfaed) and Faculty of Chemistry and Food Chemistry, Technische Universität Dresden, 01062 Dresden, Germany. [2] Helmholtz-Zentrum Dresden-Rossendorf e.V., Institute of Ion Beam Physics and Materials Research, 01328 Dresden, Germany. [3] Shanghai Synchrotron Radiation Facility, Zhangjiang Laboratory, Shanghai Advanced Research Institute, Chinese Academy of Sciences, 201204 Shanghai, China. [4] Fraunhofer Institute for Ceramic Technologies and Systems (IKTS), Maria-Reiche-Strasse 2, 01109 Dresden, Germany. [5] Department of Applied Chemistry, School of Applied and Natural Sciences, Northwestern Polytechnical University, 710129 Xi'an, China. [6] Department of Applied Physics, Aalto University, P.O. Box 11100FI-00076 Aalto, Finland. [7] These authors contributed equally: Haixia Zhong, Mahdi Ghorbani-Asl, Khoa Hoang Ly, Jichao Zhang. [8] These authors jointly supervised this work: Renhao Dong, Xinliang Feng. ✉email: xinliang.feng@tu-dresden.de

Electrocatalytic carbon dioxide reduction reaction ($CO_2RR$), coupled to renewable energies, offers sustainable opportunities towards the production of value-added chemicals and carbon-based fuels[1,2]. Specifically, the electrochemical reduction of $CO_2$ to $C_1$ products (i.e., formic acid and CO) is of high relevance for the chemical industry[3–5], which can also yield a mixture of CO as carbon-reduced product and $H_2$ as byproduct. This so-called syngas mixture with varying molar ratios of $H_2$/CO is commonly used as precursor for hydroformylation process ($H_2$: CO = 1:1), methanol synthesis of Fischer−Tropsch process ($H_2$: CO = 2:1) and methanation process ($H_2$:CO = 3:1) to produce high-value/energy-dense hydrocarbons/alcohols[5,6]. However, the $CO_2$-to-CO conversion, involving the transfer of two electrons and two protons, typically suffers from high kinetic barriers and low selectivity due to the high thermodynamic/kinetic stability of $CO_2$ and the competing hydrogen evolution reaction (HER) in aqueous media, respectively[7,8]. Recently, great efforts have been dedicated to explore numerous electrocatalysts for catalytic $CO_2RR$ to CO, including noble metals (Au, Pd, Ag), transitional-metal-based materials (Fe, Co, Ni, Cu, Zn) and heteroatom-doped porous carbons[2,7,9–17]. Nevertheless, these single site catalysts still fail to meet the requirement of electrochemical syngas synthesis with tunable $H_2$/CO ratio (from 1:1 to 3:1 or higher) at relatively low overpotential. Therefore, the development of electrocatalysts with high activity and selectivity that enables tuning the competitive reactivity between the $CO_2RR$ and HER at room temperature is imperative.

Metal-organic frameworks (MOFs) are one class of highly ordered crystalline coordination polymers, which are emerging as highly attractive catalytic systems due to the uniquely combining homogenous and heterogeneous features: First, the incorporation of well-defined and highly active sites into a defined and stable scaffold ensures excellent catalytic activity and selectivity; second, the porous metrics allow for sufficient and controllable mass transfer to and from the embedded active sites; finally, the molecularly defined catalytic environment around the active site allows for tuning the catalytic reaction by modifying the scaffold and facilitates deriving fundamental understanding of catalytic mechanism. For example, Co-PMOF and $Al_2(OH)_2TCPP$-Co MOFs have been developed for electrocatalytic $CO_2RR$ to CO with high selectivity (>70%)[8,13,18]. However, conventional MOFs suffer from certain drawbacks including their intrinsically low conductivity (electrical insulators) and the blockage of metal centers by organic ligands, which have greatly hampered their development for promoting $CO_2RR$. Recent researches have demonstrated that layered 2D conjugated MOFs (2D c-MOFs)[19–23] with fully in-plane $\pi$-delocalization along 2D directions and weak out-plane $\pi−\pi$ stacking exhibit higher density of exposed metal centers and improved electron conductivity (up to 2500 S cm$^{-1}$)[24] apart from the inherited features of traditional MOFs, suggesting a great potential in high-performance electrocatalysis. For instance, the reported THT-Ni (THT = triphenylenehexathiol)[25] and THT-Co[26] 2D c-MOFs with Ni(Co)S$_4$ active sites afford superior HER electrocatalytic activity; Ni$_3$(hexaiminotriphenylene)$_2$ 2D c-MOF with NiN$_4$ active sites[27] and phthalocyanine (Pc)-based 2D c-MOF with CoO$_4$ active sites[28] can efficiently catalyze oxygen reduction reaction with onset potential of 0.82 V and half-wave potential of 0.83 V vs. RHE (reversible hydrogen electrode) in alkaline media, respectively. Inspired by these successes, we anticipate that 2D c-MOFs should also act as promising electrocatalysts in enhancing $CO_2RR$ even though the related reports are rather limited[29]. Furthermore, the competitive reactivity between the $CO_2RR$ and HER can be presumably tuned by tailoring the structures and compositions of 2D c-MOFs. Therefore, as a proof-of-concept based on the above structural/property advantages, we rationally designed layered 2D c-MOFs with bimetallic centers to improve electrocatalytic $CO_2RR$ activity toward syngas synthesis; hereby one metal center will show high selectivity for $CO_2$-to-CO conversion while the other metal center will be utilized for $H_2$ generation due to its low binding energy of CO and high proton generation rate.

Herein, a 2D c-MOF electrocatalyst with bimetallic centers is synthesized by solvothermal approach for electrocatalytic $CO_2RR$. This 2D c-MOF consists of phthalocyaninato copper as the ligand and zinc-bis(dihydroxy) complex (ZnO$_4$) as the linkage, named as (PcCu-O$_8$-Zn). The electrochemical measurements indicate that PcCu-O$_8$-Zn exhibits highly selective catalytic activity for $CO_2$-to-CO conversion (88%) and high turnover frequency (TOF) of 0.39 s$^{-1}$ at −0.7 V vs. RHE and excellent stability. Syngas compositions with different molar $H_2$/CO ratio (from 1:7 to 4:1) can be tuned via varying the metal centers (Cu and Zn) of ligand/ linkage as well as applied potentials. Operando X-ray absorption spectroscopy (XAS) and surface-enhanced infrared absorption (SEIRA) spectroelectrochemistry are utilized to probe the catalytic sites and the reaction process. The spectroscopic studies combined with contrast experiments and density functional theory (DFT) calculation reveal that ZnO$_4$ complexes in the linkages of PcCu-O$_8$-Zn exhibit high catalytic activity for $CO_2$-to-CO conversion, while CuN$_4$ complexes in the Pc macrocycles act as the synergetic component to promote the protonation process and hydrogen generation along with the $CO_2RR$. Thus, the bimetallic active sites contribute to a synergistic effect on the $CO_2RR$. Our work highlights the bimetallic MOF electrocatalyst for highly selective $CO_2RR$.

## Results

**Material design and reaction energetics.** Density functional theory calculations were firstly employed to optimize the electrocatalyst design by simulating the reaction energetics of $CO_2RR$ and the competing HER on Pc-based 2D c-MOFs (PcM-O$_8$-M1, M=Cu or Zn, M1=Cu or Zn) (Fig. 1 and Supplementary Figs. 1–4 and Tables 1–7). Typically, the electrochemical $CO_2$-to-CO reduction steps include the first proton-coupled electron transfer to generate a carboxyl intermediate (*COOH), and subsequently the second charge transfer (one electron and one proton) for the formation of *CO intermediate, as well as the desorption of CO for the final CO product (Eqs. 1–3 in Supplementary Methods)[30,31]. On the other hand, HER goes through a proton (*H) intermediate. The results of calculations reveal that the formation of *COOH via protonation is the rate-limiting step for PcM-O$_8$-M1 (Fig. 1a, c and Supplementary Figs. 3, 4). The calculated binding energy values of the intermediates (*COOH) and *H on PcM-O$_8$-M1 manifest a stronger interaction of *COOH intermediate and a weaker interplay of *H with the linkages (M1O$_4$ complexes) as compared with those of the phthalocyanine macrocycles (MN$_4$ complexes, Supplementary Tables 2–5). It is thus proposed that M1O$_4$ and MN$_4$ complexes serve as the catalytic sites for $CO_2RR$ and HER, respectively. On the other hand, in the $CO_2RR$ process, the ZnO$_4$ complexes of PcCu-O$_8$-Zn display the lowest Gibbs free energy for *COOH formation and the lowest overpotential compared to other M1O$_4$ complexes in PcM-O$_8$-M1 (Fig. 1a, c and Supplementary Figs. 3, 4 and Supplementary Table 6), suggesting that the electrochemical $CO_2RR$ to CO is energetically preferred for PcCu-O$_8$-Zn. In addition, the overpotential for $CO_2RR$ at M1O$_4$ (Supplementary Table 6) has been found to be influenced by different MN$_4$ complexes in the Pc ligand. For example, the overpotential of PcCu-O$_8$-Zn is lower than that of PcZn-O$_8$-Zn (while PcCu-O$_8$-Cu < PcZn-O$_8$-Cu), which reveals the important role of the Pc metal centers on promoting electrocatalytic $CO_2RR$. To achieve more insight into the role of Pc metal centers, we further compare the free energy profiles of HER on MN$_4$ and

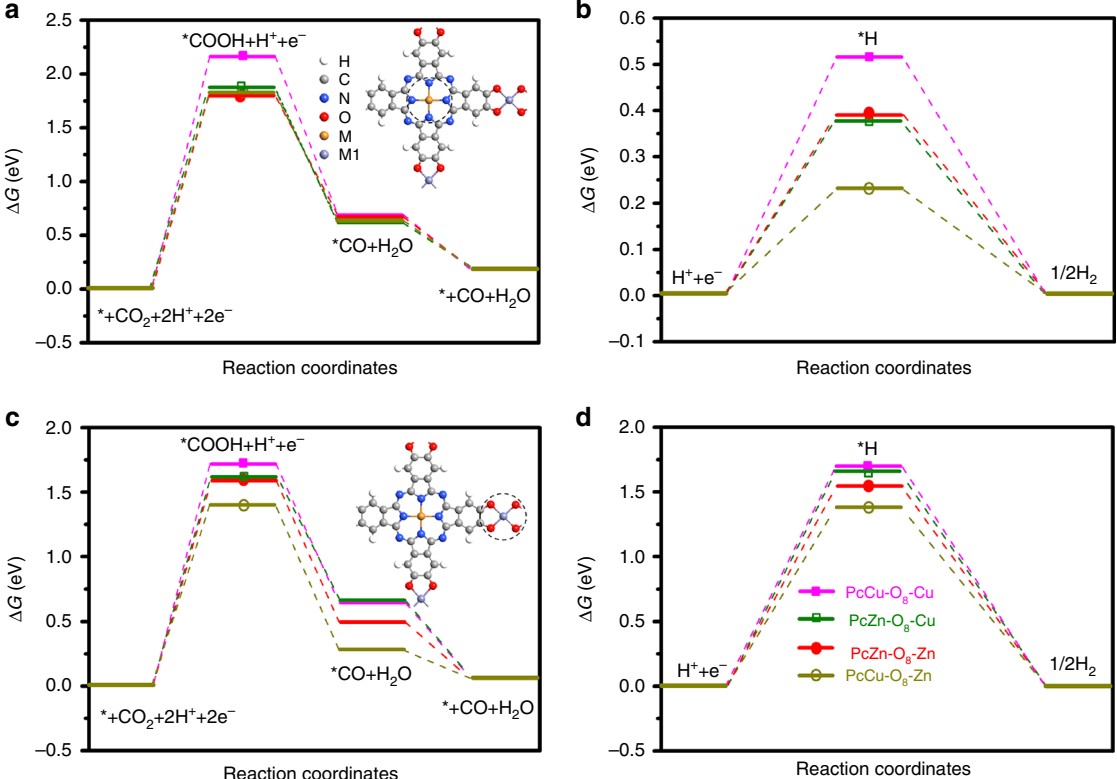

**Fig. 1 DFT calculation of CO$_2$RR. a, b** Free energy profiles of CO$_2$RR and HER on MN$_4$ units of PcM-O$_8$-M1 at $U = 0.0$ V, respectively. Inset in (**a**): atomistic structure of PcM-O$_8$-M1. The dashed circle indicates the catalytic active area with MN$_4$ units. **c, d** Free energy profiles of CO$_2$RR and HER on M1O$_4$ units in the PcM-O$_8$-M1 at $U = 0.0$ V, respectively. Inset in (**c**): atomistic structure of PcM-O$_8$-M1. The dashed circle indicates the catalytic active area with M1O$_4$ units.

M1O$_4$ complexes, which show that CuN$_4$ complex in PcCu-O$_8$-Zn exhibits the lowest HER energy barrier and the fastest proton/ electron transfer kinetics among the different metal centers (Fig. 1b, d, Supplementary Table 7)[32,33]. Based on the above consideration, it is inferred that the presence of CuN$_4$ complexes in the PcCu-O$_8$-Zn MOF facilitates the protonation of adsorbed *CO$_2$ on ZnO$_4$ complexes, and thus accelerates the overall CO$_2$RR kinetics. Therefore, a synergistic effect between CuN$_4$ complexes and ZnO$_4$ complexes is proposed for bimetallic MOF electrocatalysts.

**Synthesis and characterization.** Inspired by the above theoretical calculation, PcCu-O$_8$-Zn MOF comprising of 2,3,9,10,16,17,23,24-octahydroxy phthalocyaninato copper monomer linked by square planar ZnO$_4$ linkages (Fig. 2a) was synthesized via solvothermal method (Supplementary Figs. 5–7), as confirmed by Fourier-transform IR (FT-IR) spectroscopy and powder X-ray diffraction (XRD) measurements. The disappearance of the ligand OH signals (3300 and 630 cm$^{-1}$) and the peak shift from 1288 cm$^{-1}$ (C-OH) to 1270 cm$^{-1}$ (C-O-Zn) in the FT-IR spectra (Supplementary Fig. 8) demonstrate the successful coordination of O to Zn atoms[34]. The XRD pattern (Fig. 2b) shows intense peaks at 5.0°, 7.1° and 10.1°, assignable to (100), (110) and (200) plane, respectively, which indicates the long-range order within the *ab* plane[35]. The broad peak at 27.3° originates from the weak long-range stacking along the *c* direction with a layer distance of 0.33 nm, which is a typical feature of layered MOFs[36]. Compared to the calculated structures, the observed XRD pattern of PcCu-O$_8$-Zn is in a good agreement with the AA staggered stacking geometry. Scanning electron microscopy (SEM, Supplementary Fig. 9) images indicate aggregated nanosheets in the resulting MOF samples.

Transmission electron microscopy (TEM) images also present a mass of MOF nanosheets with an average size of 24 nm (Fig. 2c). The selected area electron diffraction pattern (SAED, inset image in Fig. 2c) and the high-resolution TEM (HR-TEM, Fig. 2d) images further manifest the crystalline structure of PcCu-O$_8$-Zn based on a square lattice of 1.75 nm.

Element mapping images (Supplementary Fig. 10) disclose the homogenous distribution of Cu, Zn, C, N and O in the PcCu-O$_8$-Zn sample. Furthermore, X-ray photoelectron spectroscopy (XPS) analysis also confirms the presence of Cu, Zn, C, N and O elements (Supplementary Fig. 11). In the high-resolution Cu *2p* spectrum, the set of peaks at 936.7 and 953.8 eV is assigned to Cu *2p$_{3/2}$* and Cu *2p$_{1/2}$*, respectively, which suggests one type of oxidized Cu (II) in the PcCu-O$_8$-Zn[37]. The deconvolution of N1s spectra further verifies the coordination of Cu and N[38]. For the high-resolution scan of the Zn *2p* region, the typical feature of Zn (II) is found[39].

To further investigate the chemical state of the Cu and Zn atoms in the PcCu-O$_8$-Zn sample, XAS and extended X-ray absorption fine structure (EXAFS) analyses were performed. The Cu *K*-edge X-ray absorption near-edge structure (XANES) spectra (Supplementary Fig. 12) show that both PcCu-O$_8$-Zn and the monomer PcCu-(OH)$_8$ exhibit a typical Cu(II) peak at 8985 eV (*1s* to *3d* electron transition), which is similar to that of the reference copper(II) phthalocyanine (PcCu), thus confirming the presence of Cu-N in PcCu-O$_8$-Zn[40,41]. Generally, two characteristic signals are observed in the Zn XANES spectra including the pre-edge peak at around 9660 eV and the main absorption peak at 9660−9680 eV, which correspond to the electron transition from *1s* to *3d* (typically found for the transition metal Zn) and the *1s* to *4p* electronic transition,

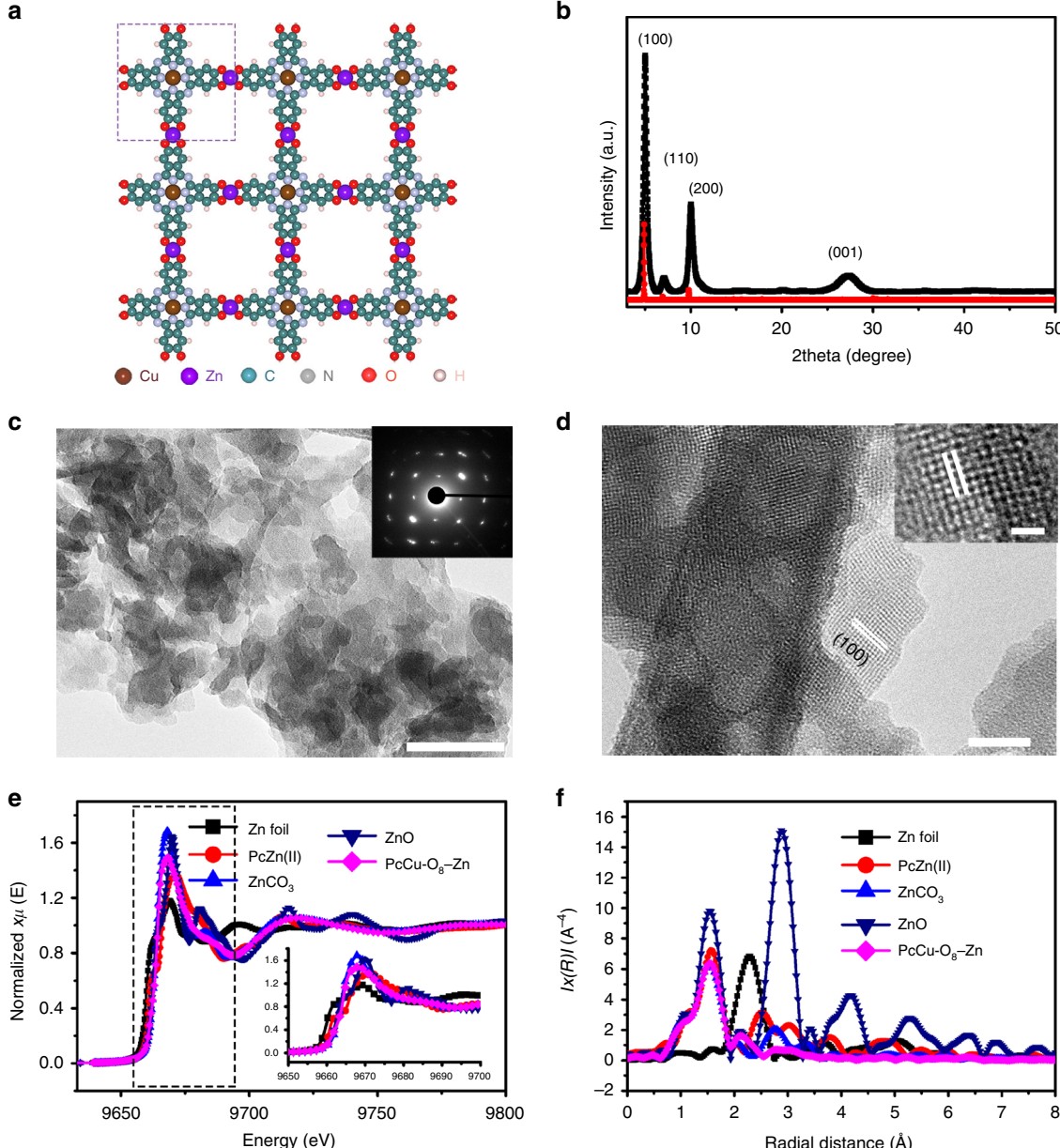

**Fig. 2 Structure and morphology of PcCu-O$_8$-Zn. a** Schematic structure of PcCu-O$_8$-Zn (the dashed rectangular indicates the unit cell). **b** Experimental (black square dot line) and calculated (red circle dot line) XRD patterns of PcCu-O$_8$-Zn. **c** TEM image of PcCu-O$_8$-Zn sample (inset: selected area electron diffraction pattern), Scale bar: 60 nm. **d** HR-TEM image of PcCu-O$_8$-Zn sample. Scale bar: 20 nm (inset: 5 nm). **e** Zn $K$-edge XANES spectra for Zn foil, ZnO, ZnCO$_3$, PcZn(II) and PcCu-O$_8$-Zn. Inset: the enlarged figure of the XANES spectra between 9650 and 9700 eV. **f** Fourier transform EXAFS of Zn foil, ZnO, ZnCO$_3$, PcZn and PcCu-O$_8$-Zn.

respectively[42]. As shown in Fig. 2e, compared to Zn foil, the pre-edge peak signal at 9660 eV is not detected in the Zn $K$-edge spectrum of PcCu-O$_8$-Zn due to the full occupied $3d$ orbital of Zn$^{2+}$, therefore excluding the existence of Zn(0) in PcCu-O$_8$-Zn. In addition, PcCu-O$_8$-Zn also shows a main peak at 9665 eV similar to the ZnCO$_3$ and ZnO (Fig. 2e), which suggests the oxidation valence of Zn atom as +2 in PcCu-O$_8$-Zn[42]. Figure 2f displays the radial structure functions of PcCu-O$_8$-Zn and clearly demonstrates the characteristic Zn-O coordination in PcCu-O$_8$-Zn with intensive peak at around 1.55 Å. The absence of obvious structural peaks and the diminishment of the signal at 2.27 Å in Fig. 2f reveal that no heavy backscattering atoms (Zn) are bound to Zn sites in PcCu-O$_8$-Zn[43,44]. Therefore, the XANES and EXAFS spectra of PcCu-O$_8$-Zn together with the contrast experiments provide solid proof for the existence of square

planar complexes via the coordination of PcCu(II)-(OH)$_8$ to Zn (II) ions.

Low-pressure N$_2$ sorption was measured to evaluate the porous properties of PcCu-O$_8$-Zn (Supplementary Fig. 13). The Brunauer Emmett Teller surface area was measured to be 378 m$^2$ g$^{-1}$. The pore size distribution indicates its abundant micropores (1.4 nm) and mesopores (6 nm), which can be favorable for the mass transport during the catalytic process[28].

**CO$_2$RR activity evaluation.** The electrocatalytic CO$_2$RR activity of PcCu-O$_8$-Zn was evaluated in a two-compartment electro-chemical cell in 0.1 M KHCO$_3$ aqueous electrolyte. The PcCu-O$_8$-Zn/carbon nanotube (CNT) composite with a weight ratio of 2:1 (details provided in Methods section) was loaded on carbon

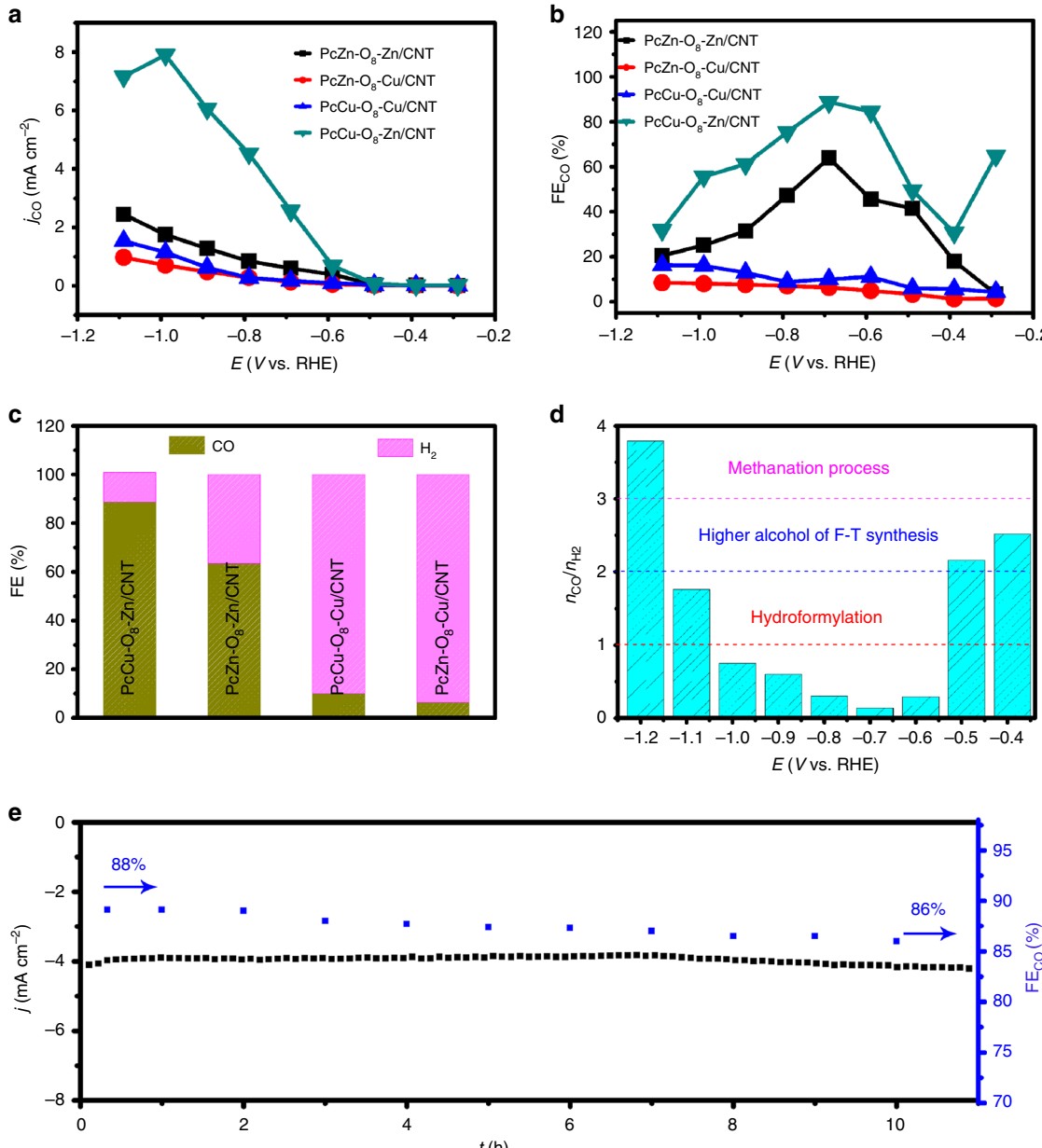

**Fig. 3 CO2RR performance. a** Partial current and **b** Faradaic efficiency of CO for PcCu-O8-Zn/CNT, PcCu-O8-Cu/CNT, PcZn-O8-Zn/CNT and PcZn-O8-Cu/CNT at different potentials. **c** Faradaic efficiency of CO and H2 for PcCu-O8-Zn/CNT, PcCu-O8-Cu/CNT, PcZn-O8-Zn/CNT and PcZn-O8-Cu/CNT at −0.7 V vs. RHE. **d** Molar H2/CO ratio at different applied potentials of PcCu-O8-Zn/CNT. **e** Amperometry ($i$ ~ $t$) stability and the according Faradaic efficiency for CO of PcCu-O8-Zn/CNT at −0.7 V vs. RHE in CO2-saturated 0.1 M KHCO3.

paper, which was contacted as the working electrode. The contrast composite samples of 2D $c$-MOFs/CNT (PcZn-O8-Zn/CNT, PcZn-O8-Cu/CNT and PcCu-O8-Cu/CNT) were also synthesized (details in Supplementary Information, Supplementary Figs. 14–17) and evaluated for electrocatalytic CO2RR activity. Cyclic voltammetry (CV) experiments revealed increased current densities for PcCu-O8-Zn/CNT in CO2-saturated media compared to Ar-saturated electrolyte (Supplementary Fig. 18), demonstrating its feasibility for CO2RR. To verify that the currents originate from the catalytic CO2RR, constant potential electrolysis was performed. The products were detected via gas chromatography (GC) and nuclear magnetic resonance (NMR) measurements. The results indicate that only gaseous (H2 and CO) products were generated at the applied potentials with total Faradaic efficiency of 99 ± 2.2% (Supplementary Figs. 19–21). The resultant CO2RR

catalytic performance including the maximum CO efficiency and the molar CO/H2 ratio suggested strong dependence on the type of metal centers and applied potential (Fig. 3a, b and Supplementary Fig. 20). Among the synthesized 2D c-MOFs/CNT hybrids, the PcCu-O8-Zn/CNT sample yielded the highest partial current density for CO ($j_{CO}$) and the highest corresponding Faradic efficiency toward CO (FE$_{co}$) over the investigated potential range (Fig. 3a, b), indicating superiority of the ZnO4 sites for selective conversion of CO2 to CO over ZnN4, CuN4 and CuO4 centers, which is also supported by the DFT calculations (Fig. 1). Notably, $j_{CO}$ for PcCu-O8-Zn/CNT showed a maximum value at −1.0 V vs. RHE, while H2 generation ($j_{H2}$) displayed a steady rise with the increased overpotential (Supplementary Fig. 20a). This observation can be attributed to the competitive reactivity between the CO2RR and HER as well as the limitation

of the transport of $CO_2$ to the catalytic sites[45]. At $-0.7$ V vs. RHE, the $FE_{CO}$ for PcCu-$O_8$-Zn/CNT reached up to 88%, which is significantly higher than that of PcZn-$O_8$-Zn/CNT (63%), PcZn-$O_8$-Cu/CNT (6%), and PcCu-$O_8$-Cu/CNT (10%) and the other reported Zn- and MOF-based electrocatalysts (up to ~80%, seen in Supplementary Table 8)[13,18]. Although $CuN_4$ is more efficient for HER compared to $ZnN_4$ based on theoretical calculations, PcCu-$O_8$-Zn/CNT consisting of $CuN_4$ and $ZnO_4$ complexes still shows higher $FE_{CO}$ than that of PcZn-$O_8$-Zn/CNT comprising $ZnN_4$ and $ZnO_4$ complexes. This points to the synergistic catalytic effects between $CuN_4$ and $ZnO_4$ in selective electroreduction of $CO_2$ to CO. In addition, PcCu-$O_8$-Zn/CNT also shows a high TOF of $0.39 s^{-1}$, which is superior to other MOFs and Zn-related electrocatalysts (Supplementary Table 8). The molar ratio of the syngas $CO/H_2$ catalytically generated by the 2D $c$-MOFs could be additionally controlled via the applied potentials. As shown in Fig. 3d, the molar $H_2/CO$ ratio for the PcCu-$O_8$-Zn/CNT system could be tuned from around 1:7 to 4:1 by increasing the applied potential from $-0.4$ to $-1.2$ V vs. RHE.

To elucidate the kinetics of these MOFs toward the catalytic $CO_2RR$, Tafel slopes were derived (Supplementary Fig. 22a). PcCu-$O_8$-Zn/CNT exhibited the lowest Tafel slope ($125 mV dec^{-1}$) toward the CO production compared to PcZn-$O_8$-Zn/CNT (145 $mV dec^{-1}$), PcZn-$O_8$-Cu/CNT ($223 mV dec^{-1}$) and PcCu-$O_8$-Cu/ CNT ($280 mV dec^{-1}$), confirming its faster kinetics. Electrochemical impedance spectroscopy results indicate that PcCu-$O_8$-Zn/ CNT exhibits smaller resistance and faster electron kinetics than those of other as-synthesized 2D $c$-MOFs/CNT samples (Supplementary Fig. 22b). Besides, the PcCu-$O_8$-Zn/CNT system presents long-term catalytic durability. The high $FE_{CO}$ (86%) and current density were maintained over the course of 10 h of operation at $-0.7$ V vs. RHE (Fig. 3e). No obvious changes of morphology and structure (Supplementary Fig. 23) were observed in SEM image, XRD pattern, Raman and FR-IR spectra of PcCu-$O_8$-Zn/CNT after the $CO_2RR$ long-term testing, demonstrating the high stability of PcCu-$O_8$-Zn/CNT during electrocatalytic $CO_2$ conversion.

**Unveiling the active sites.** *Operando* XAS measurement was employed to gain insight into the valence state and coordination structure of Cu and Zn in the PcCu-$O_8$-Zn/CNT under the $CO_2RR$ turnover condition (Fig. 4a–d and Supplementary Fig. 24). As shown in Fig. 4a, a typical pre-edge signal of Zn(0) at around 9660 eV is not observed in the Zn $K$-edge XANES spectra for all PcCu-$O_8$-Zn/CNT samples[42,43]. This excludes the generation of metallic Zn in PcCu-$O_8$-Zn/CNT electrocatalyst during the $CO_2RR$ process. Importantly, the main absorption peak at 9665 eV was not shifted in the Zn $K$-edge XANES spectra of PcCu-$O_8$-Zn/CNT (Fig. 4a) as the applied potential was decreased to $-0.4$ (red circle dot curve) and $-0.7$ V (blue diamond dot curve) vs. RHE, respectively, and then increased back (indigo triangle dot curve) to the initial (black square dot curve) open circuit voltage (OCV). The results reveal that the oxidation state of Zn(II) in PcCu-$O_8$-Zn/CNT was maintained throughout the catalytic process, which can be explained by the fact that the Zn(II) already has a full $3d$ electron shell[46]. In addition, the pre-edge peak at 8985 eV and the main absorption peak at 8998 eV in the Cu $K$-edge XANES spectra of PcCu-$O_8$-Zn/CNT were not varied upon changing the applied potential, which indicates that the valence state of Cu(II) was not changed during the $CO_2RR$ process. Notably, the missing pre-edge peak at 8980 eV in Cu $K$-edge XANES spectra of all PcCu-$O_8$-Zn/CNT samples further confirms that no metallic Cu was generated at the PcCu-$O_8$-Zn/ CNT electrode under electrolysis condition (Fig. 4b).

To monitor the local coordination environment changes, in situ EXAFS measurements were performed. As the applied potential was performed for one cycle, the peak at 1.55 Å assigned as Zn-O bond length in PcCu-$O_8$-Zn/CNT was not shifted (Fig. 4c). Meanwhile, the peak intensity presents a negligible decrease (black square dot and indigo triangle dot curves in Fig. 4c), which is possibly due to the interaction of the reaction intermediates and the $ZnO_4$ sites during the catalytic process, such as *H, *COOH, *CO and so on[43]. Therefore, the above in situ EXAFS results reveal no obvious change in Zn coordination number and bond length of Zn-O for PcCu-$O_8$-Zn/CNT under the electrolysis condition. Furthermore, the characteristic signal of Zn$-$Zn bonding at 2.27 Å does not appear in the EXAFS spectra of all the PcCu-$O_8$-Zn/CNT samples, again excluding the formation of metallic Zn or Zn cluster at PcCu-$O_8$-Zn/CNT catalyst throughout $CO_2RR$ process. Regarding the $CuN_4$ complexes, no obvious change of the Cu-N coordination peak at 1.54 Å was detected in the Cu $K$-edge EXAFS spectra of PcCu-$O_8$-Zn/CNT (Fig. 4d) upon performing the potential in one cycle. Additionally, no obvious signal of Cu $-$Cu bonds was observed at 2.23 Å, which demonstrates that no heavy backscattering atoms (Cu) are bound to Cu sites in all PcCu-$O_8$-Zn samples. Therefore, the *operando* XAS results fully prove that the well-defined sites ($ZnO_4$ and $CuN_4$) act as stable catalytic centers during the $CO_2RR$ process, while no metals or metal clusters form via the reduction of high-valence metal centers.

Next, *operando* SEIRA spectroelectrochemistry was employed to elucidate the electrocatalytic mechanism of the 2D $c$-MOF catalysts. The 2D c-MOFs were evenly deposited as a closed film onto a nanostructured Au surface, which acted as IR signal amplifier. SEIRA spectra were recorded at different potentials covering a broad potential window. SEIRA difference spectra taken under turnover conditions were derived using the spectrum of the respective system at $-0.6$ V vs. Ag/AgCl (Fig. 4e, f). The SEIRA difference spectra of PcZn-$O_8$-Cu/CNT and PcCu-$O_8$-Zn/ CNT show distinct features that likely arise from their intrinsically different reactivities (Supplementary Fig. 25). Upon lowering the potential, a negative band at $2343 cm^{-1}$ assigned to dissolved $CO_2(g)$ was observed. This band was found to decrease with decreased potential indicating the consumption of $CO_2$ near the surface in the catalytic process[45]. Strong positive bands in the region of $1660–1640 cm^{-1}$ were observed in both cases and attributed to the changes of the interfacial $H_2O$, which accumulated in the MOFs due to catalysis or increasing negative polarization of the electrode. The high-frequency bands above $1800 cm^{-1}$ typically arise from metal bound species. Specifically, the bands located in the higher frequency region at 1933 and $2071 cm^{-1}$ were assigned to CO bound to the $CuN_4$ and $CuO_4$ centers, respectively (Fig. 4e, f)[47]. The shift of the $\nu(CO)$ mode could arise from the different electronic properties of Cu metal in the $N_4$ and $O_4$ frame, respectively. In this respect, $CuN_4$ centers can stabilize the CO via $\pi$ backbonding leading to drastically lowered $\nu(CO)$s, while CO bound to Cu and oxide-derived Cu surfaces has been reported above $2000 cm^{-1}$[47,48]. The strong band centered at $1851 cm^{-1}$ for the PcZn-$O_8$-Cu/CNT system matches the frequency for (isolated) Cu-H and is thus assigned to the Cu-H intermediate formed at the $CuO_4$ nodes in the HER cascade[47,49]. The particularly high intensity of this band suggests a dominating HER process over $CO_2RR$ at PcZn-$O_8$-Cu/CNT in $CO_2$-saturated solution. This interpretation is consistent with the electrocatalytic results (Supplementary Fig. 20), revealing that the PcZn-$O_8$-Cu/CNT system shows high selectivity for $H_2$ (>90%) over the complete potential range in $CO_2$-saturated electrolyte. In contrast, Cu-H is not observed at the $CuN_4$ units of the PcCu-$O_8$-Zn/CNT systems. This may be due to low accumulation of the Cu-H species during catalysis, which could result from the fast proton transfer kinetics at $CuN_4$ complexes to $ZnO_4$ sites and

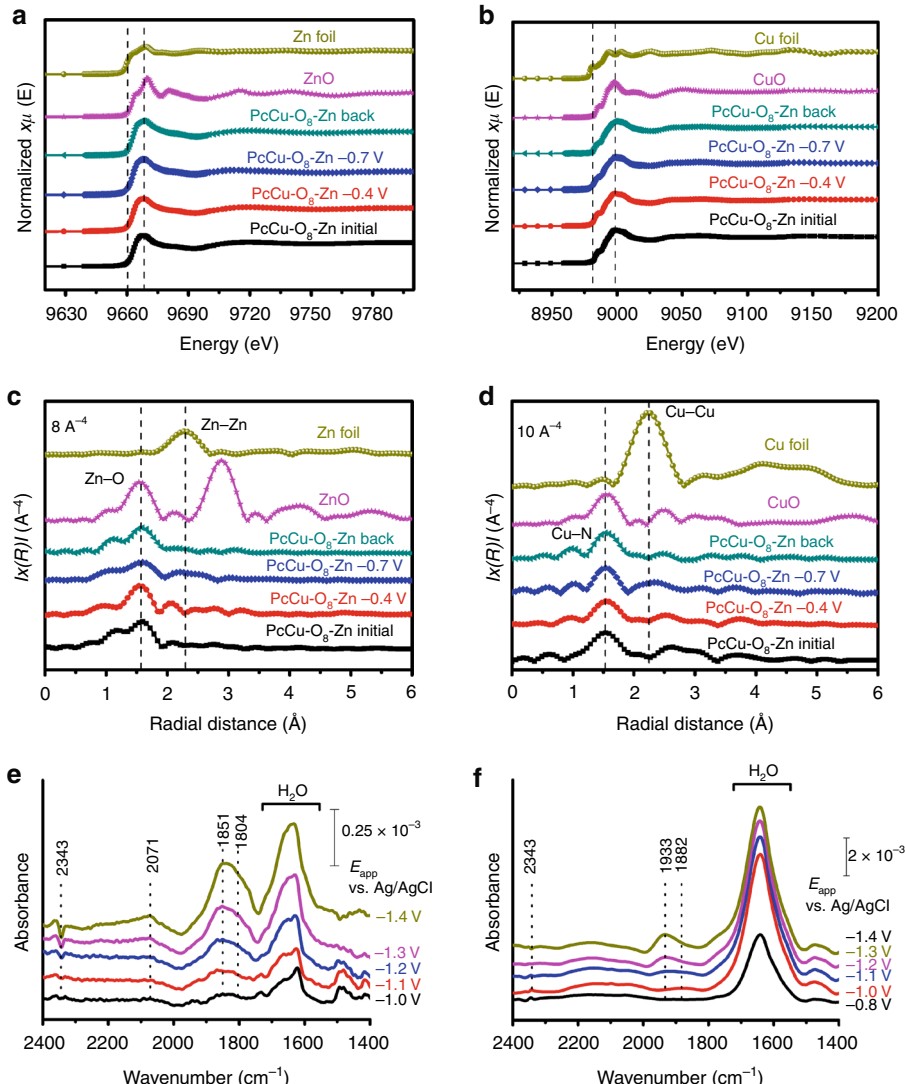

**Fig. 4** *Operando* **XAS and SEIRA measurement under electrolysis. a** Zn *K*-edge XANES spectra of Zn foil, ZnO, PcCu-O$_8$-Zn samples. **b** Cu *K*-edge XANES spectra of Cu foil, CuO, PcCu-O$_8$-Zn samples. **c** Zn *K*-edge Fourier transform EXAFS spectra of Zn foil, ZnO and PcCu-O$_8$-Zn samples. **d** Cu *K*-edge Fourier transform EXAFS spectra of Cu foil, CuO and PcCu-O$_8$-Zn samples. **e**, **f** *Operando* SEIRA spectro-electrochemical analysis of PcZn-O$_8$-Cu/CNT (**e**) and PcCu-O$_8$-Zn/CNT (**f**) in CO$_2$-saturated 0.1 M KHCO$_3$. As reference, a SEIRA spectrum of the system at −0.6 V vs. Ag/AgCl was taken.

yield H$_2$. Interestingly, no indication for CO binding to the ZnO$_4$ nodes was found due to its too low transient concentration to be observed with our current SEIRA spectro-electrochemical setup. This can be explained by the weak binding energy between ZnO$_4$ and CO, which could facilitate a quick deliberation of the product and thus suggests fast CO$_2$RR kinetics at the ZnO$_4$ complexes in PcCu-O$_8$-Zn/CNT.

## Discussion

To obtain further insight into the reactivity of 2D *c*-MOFs towards HER and CO$_2$RR, the calculated free energy profiles on M1O$_4$ site at $U = 0.55$ V were analyzed (Fig. 5a). For HER, the Gibbs free energy values of the key intermediates (*H) on M1O$_4$ units are positive, with a minimum barrier of 0.7 eV, and therefore expected to be kinetically prohibited. However, the free energy values of CO$_2$RR at the same equilibrium potential are negative, which reveals that the CO$_2$RR at M1O$_4$ site is thermodynamically downhill. It further verifies the favorable CO$_2$RR process at M1O$_4$ complexes of 2D *c*-MOFs. Although the CuN$_4$ complexes show the lowest energy barriers for HER, PcCu-O$_8$-Zn still exhibits the lowest free energy for the generation of

rate-determining *COOH intermediate as compared to the other 2D *c*-MOFs during CO$_2$RR catalysis. This establishes the synergetic effect of CuN$_4$ and ZnO$_4$ in enhancing the CO$_2$RR activity. A proposed synergistic catalytic scheme is presented in Fig. 5b. CuN$_4$ complexes attract numerous electrons and H$_2$O toward producing abundant protons, wherein protons are partially transformed into molecular H$_2$ and partially transferred to ZnO$_4$ complexes. Simultaneously, the adsorbed CO$_2$ on ZnO$_4$ complexes is reduced to *COOH by coupling with these protons/ electrons from the CuN$_4$ sites and electrode/electrolyte, and subsequently the resultant *COOH will be transformed into *CO intermediate by a further charge transfer step (one electron and one proton). The desorption of *CO results in the final CO product. As a result, the kinetics of CO$_2$RR on ZnO$_4$ is greatly enhanced in PcCu-O$_8$-Zn 2D *c*-MOF.

In summary, we have synthesized a layered 2D *c*-MOF (PcCu-O$_8$-Zn) with bimetallic centers (ZnO$_4$/CuN$_4$) capable of synergistic electroreduction of CO$_2$ to CO based on the theory-guided design. The electrocatalytic results indicated that PcCu-O$_8$-Zn mixing with CNTs exhibited high CO$_2$RR catalytic activity with high selectivity for CO conversion of 88%, TOF of 0.39 s$^{-1}$ and

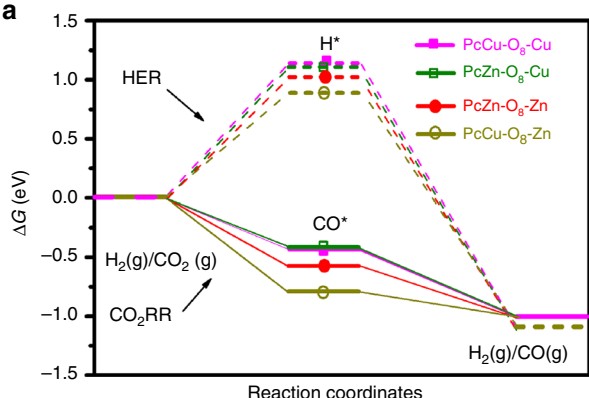

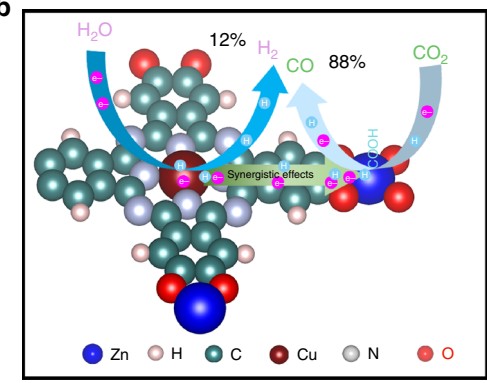

**Fig. 5 Reaction mechanism analysis. a** Free energy profiles for HER (dashed lines) and $CO_2RR$ (solid lines) on $M1O_4$ complexes of $PcCu\text{-}O_8\text{-}Zn$ at $U =$ 0.55 V. **b** Schematic HER and $CO_2RR$ reaction process of $PcCu\text{-}O_8\text{-}Zn$.

long-term durability (>10 h), which is superior to the reported MOF- and Zn-based electrocatalysts. The molar $H_2/CO$ ratio could be rationally adjusted through varying the metal centers and applied catalytic potentials, beneficial for industrial applications. Theoretical calculation and the *operando* XAS and SEIRA analysis, as well as the control experiments suggested that the $CO_2RR$ takes place at the $ZnO_4$ units while the $CuN_4$ units promote the proton and electron transfer during the reaction process. Thus, the combination of $ZnO_4$ and $CuN_4$ complexes generates a synergetic effect, which contributes to the high $CO_2RR$ performance of $PcCu\text{-}O_8\text{-}Zn/CNT$. Our work demonstrates the capability of bimetallic 2D *c*-MOFs as highly efficient electrocatalysts for promoting the $CO_2RR$, which is of importance for conductive MOFs design and their electrocatalysis application and also sheds light on the development of high-performance bimetal-heteroatom doped carbon electrocatalysts.

## Methods

**Computational studies**. The computational modeling of the reactants, intermediates and products, and reaction process involved in the reactions on 2D MOFs was performed by using DFT with the PBE exchange-correlation functional[50], as implemented in the VASP code[51,52]. The total energies were converged within $10^{-6}$ eV/cell. The cut-off energy for plane wave basis was set at 500 eV. The Brillouin zone of the supercells was sampled using $4 \times 4 \times 1$ Monkhorst–Pack grid of *k*-points. All calculations have been performed using the spin-polarized setup. Dispersion interactions were taken into account as proposed by Grimme within the DFT-D2 scheme[53]. More detail and relevant reference are provided in the Supplementary Information.

**Synthesis of PcCu-$O_8$-Zn (Cu)**. PcCu-$(OH)_8$ (0.005 mmol) was well dispersed into DMF (N,N-Dimethyformamide, 25 ml). After adding 2.2 ml of $NH_3H_2O$ (50%), the mixture solution was diluted by 30 ml of $H_2O$. After sonication for 5 min, 5 ml of $M(acac)_2$ (M=Cu, Zn, 0.01 mmol) solution was added into the above mixture. This mixture was transferred into the autoclave and heated at 120 °C for 40 h. Finally, the solid was obtained by filtration, washing with DMF, $H_2O$ and acetone, and vacuum drying at room temperature. PcM-$O_8$-M1/CNT (M, M1=Cu or Zn) was synthesized with the similar procedure except that 15 mg of CNT was mixed with the PcCu-$(OH)_8$ precursor, wherein the ratio of PcM-$O_8$-M1 and CNT is around 2:1.

**Characterization**. Powder XRD measurements were collected on a PW1820 powder diffractometer (Phillips) using Cu-Kα radiation ($\lambda = 0.15418$ nm, 40 kV, 30 mA). TEM images were obtained using a Cs-corrected TEM (Carl Zeiss Libra 200) operated at 200 kV. SEM was recorded on Zeiss Gemini S4 500. Raman spectra were collected with a Renishaw 2000 model confocal microscopy Raman spectrometer with a CCD detector and a holographic notch filter at ambient conditions. FT-IR tests were performed on a Bruker Optics ALPHA-E spectrometer equipped with Attenuated Total Reflectance (ATR) sample holder. The porosity was detected by nitrogen sorption using a micromeritics ASAP 2020 analyzer. XPS spectra were collected with an ESCALAB MK II X-ray photoelectron spectrometer using an Al Kα source. The rotating disk electrode (RDE) was performed on MSR electrode rotator (Pine Instrument Co.). The XAS and EXAFS data

were collected at room temperature in transmission mode at beamline BL14W1 and BL15U1 of the Shanghai Synchrotron Radiation Facility (SSRF, China).

**Electrode preparation**. One milligram of catalyst was added into 100 μl of ethanol containing 10 μl of Nafion solution (5% in ethanol) and ultrasonically treated for 30 min. And the catalyst ink was drop-casted onto carbon paper.

**Electrochemical test**. Before testing, the Nafion membrane (115) was treated in $H_2O_2$ solution (5%) and pure water for 1 h. And the carbon paper with loading catalyst, Pt mesh and Ag/AgCl are used as the working, counter and reference electrode. Firstly, the electrolyte in the cathodic compartment was degassed by bubbling with Ar for at least 30 min for removal of oxygen, and then purged continuously with $CO_2$. $CO_2$ gas was delivered into the cathodic compartment at a rate of 30.00 sccm and was vented directly into the gas-sampling loop of a gas chromatograph. GC run was initiated every 20 min. All reference electrodes are converted to the RHE reference scale using $E$ (vs. RHE) = $E$ (vs. Ag/AgCl) + 0.197 V + 0.0591 V × pH.

The partial current densities of CO and $H_2$ production were calculated from the GC peak areas as follows:

$$j_{CO/H_2} = v_{CO/H_2} \times \text{flow rate} \times \frac{2FP_0}{RT} A^{-1}, \quad (1)$$

where $V_{CO}$ and $V_{H_2}$ are the volume concentration of CO and $H_2$, respectively, $P_0$ is the standard atmospheric pressure (1.013 bar), $T$ is the absolute temperature (273.15 K), $F$ is Faradaic constant (96,485 C mol$^{-1}$), and $A$ is the electrode area (1 cm$^2$). Faradaic efficiencies for a given product were calculated by dividing these partial current densities by the total current density.

The liquid products were analyzed by NMR spectroscopy, in which 0.5 ml of the electrolyte was mixed with 0.1 ml $D_2O$ and 0.05 μl dimethyl sulfoxide (DMSO), wherein DMSO was serviced as an internal standard. The one-dimensional $^1H$ spectrum was measured with water suppression using a pre-saturation method.

**Operando XAS measurement**. Operando XANES and EXAFS experiments were carried out at the BL14W1 beamline of the Shanghai Synchrotron Radiation Facility (SSRF). All data were collected in fluorescence mode under applied potential controlled by CHI electrochemical workstation. A custom-designed cell (Supplementary Fig. 21) was used for the in situ XAS measurements, which was applied to the identical conditions as the real $CO_2RR$ testing. The X-ray energy was calibrated using a Cu metal foil and Zn metal foil.

**Operando SEIRA spectro-electrochemistry**. All measurements were conducted in aqueous $CO_2$ saturated 0.1 M $KHCO_3$. An FT-IR spectrometer (*Bruker* IFSv66) equipped with a $N_2$-cooled MCT detector was employed. The measurements were carried out in attenuated total reflection (ATR) mode in Kretschmann geometry using an Si prism as IR active waveguide. A thin and nano-scale rough Au layer was coated onto the prism for conductivity/contacting purposes prior to MOF deposition/drop-casting. Deposition of the Au film is described elsewhere[54]. MOF drop-casting followed procedures as described above. The Au layer acted as a signal amplifier giving rise to strong surface-enhancement of IR signals of compounds close to the Au surface. In this way, we achieve to record SEIRA spectra of the MOF layers close to the electrode surface, which should exhibit excellent electronic contact. For applying potentials, the MOF-coated prism was mounted into a customized three-electrode containing spectro-electrochemical cell as described elsewhere[54]. A hydrogen-flamed cleaned Pt wire and Ag/AgCl in 3 M KCl (DriRef, World Precision Instruments) acted as counter and reference electrode, respectively.

## Data availability

The datasets generated during and/or analyzed during the current study are available from the corresponding author on reasonable request.

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

## Acknowledgements

This work is financially supported by EU Graphene Flagship (GrapheneCore2 785219) and Coordination Networks: Building Blocks for Functional Systems (SPP 1928, COORNET), as well as the German Science Council and Center of Advancing

Electronics Dresden (cfaed). This project also has received funding from the European Research Council (ERC) under the European Union's Horizon 2020 research and innovation program (FC2DMOF, grant agreement No. 852909). H.Z. gratefully acknowledges funding from the Alexander von Humboldt Foundation. I.M.W. acknowledges the Cluster of Excellence UniSysCat. We acknowledge Dresden Center for Nanoanalysis (DCN) at TUD and Dr. Petr Formanek (Leibniz Institute for Polymer Research, IPF, Dresden) for the use of facilities. We also thank Mr. Zhiyong Wang and Dr. Chongqing Yang for helpful discussions. We thank the beamline scientists at BL14W1 and BL15U1 of the Shanghai Synchrotron Radiation Facility for the XAFS measurements. We thank Mr. Chenbao Lu and Prof. Xiaodong Zhuang for the in-situ XAS electrochemical cell setup, as well as Wei Li for the ex-situ Raman testing. We thank Mr. Zhe Zhang for helping prepare the Au/CP current collector. The computational support from the HZDR computing cluster is gratefully appreciated.

## Author contributions

R.D. and X.F. conceived and designed the project. H.Z. synthesized the precursor and MOFs and conducted the morphology, structural, compositional and electrochemical $CO_2RR$ performance. M.G.-A. and A.V.K. contributed to the theoretical calculations. K.H.L. and I.M.W. contributed to the in-situ FT-IR measurements and analysis. J.G. and D.M. help with the $CO_2RR$ performance testing. Jichao Z. conducted Operando and ex-situ XAS and EXAFS measurements and related analysis. M.W. contributed to the ligand's synthesis. Z.L. and E.Z. contributed to the TEM testing. Jian Z., E.B. and S.K. contributed to the discussion of the MOFs synthesis and electrochemical performance. H.Z., R.D., K.H.L. and X.F. co-wrote the paper. All the authors discussed the results and commented on the manuscript.

## Competing interests

The authors declare no competing interests.
