## [Peer Review File · Nature Communications]

Reviewers' comments:

Reviewer #1 (Remarks to the Author):

The authors report bimetallic two-dimensional conjugated metal-organic frameworks (2D c-MOF) applied for CO₂ reduction reaction (CO₂RR) electrocatalysts.

They made 2D c-MOFs which spatially disperse single-metal-atom sites with different coordinate structures such as N-coordinated or O-coordinated sites in a single MOF. Specifically, their 2D c-MOF is composed of Cu-phthalocyanine as a ligand (CuN₄) of MOF and Zn-bis(dihydroxy) complex (ZnO₄) as a linkage of MOF. They sought to study the role of each individual atomic site as an active site for CO₂RR and hydrogen evolution reactions (HER) by matching trends in CO₂RR performance, DFT calculations, and operando surface-enhanced infrared-absorption spectroelectrochemistry (SEIRA). As the interest in single atom catalysts (SACs) is growing in electrocatalysis, this approach is of potential interest.

However, current manuscript has a serious problem because the exact nature of the material (especially 2D c-MOF) during CO₂RR has not been studied properly. This means that the DFT calculations and operando SEIRA (especially for intermediate detection) do not support the mechanistic proposal.

Also, the CO₂RR performance is not impressive.

Here are further comments:

1. Authors defined TOF as 'turn of frequency' in both abstract (page 2) and introduction (page 4). TOF is 'turnover frequency', not 'turn of frequency'.
2. Authors proceeded the DFT calculation (Figure 1, Figure 5) by assuming the structure of isolated Cu atom and isolated Zn atom in their 2D c-MOF. However, they didn't provide information about the real structure of 2D c-MOF during CO₂RR. We need to know whether the MOF is stable or not during CO₂RR. In the experience of most researchers, it is not, i.e. the MOF changes dramatically in electrolyte and under reducing conditions, especially over extended operating conditions. Although they reported the SEM and XRD (Figure S18) after CO₂RR, this is not enough to know the real structure of the material.
 - 2-1. Operando XAS experiment during CO₂RR is required for both Cu K edge and Zn K edge. Their XAS result (Fig. 2e, f, Fig. only shows the result of as-synthesized 2D c-MOF. Especially it has been reported in several papers that Cu ion (coordinated in N and O) is reduced to Cu (0) and Cu clusters form from organometallic complex and MOF during CO₂RR. [1,2]
 - 2-2. In operando EXAFS data, the coordination number and atomic distance of coordinated atom around Cu and Zn need to be provided by EXAFS fitting. This result is important if they want to prove the N-coordinated, O-coordinated metal atomic site during CO₂RR.
 - 2-3. In XANES data, the magnified XANES spectra at the edge energy region need to be provide. In current XANES (Figure 2e), it is hard to distinguish the XANES positions according to samples.
 - 2-4. FT-IR or Raman study during CO₂RR or after CO₂RR is required to prove the stability of MOF (especially organic materials).
3. For the CO₂RR stability measurement (Fig. 3e), more CO FE data points are required between start

and end point.

4. When compare the CO₂RR performance with other MOFs related materials, current density, potential should be provided in the table (Table S8). Also, it seems that not all the results of MOFs based CO₂RR electrocatalyst are not included in the table. For example, a paper that reported the Co porphyrin-based COF for CO₂RR (90% CO FE, 24 h stability) has not been included in the table. [3]

5. In this work, operando SEIRA analysis was done with 2D c-MOFs deposited onto a nanostructured Au surface. Although the authors mentioned 2D c-MOFs as a 'closed film', Au underlayer can affect the CO₂RR because Au is CO₂RR catalyst. 2D c-MOFs seem not fully continuous to cover all the surface area when I see the SEM images (Figure S6, S11, S12, S13). Therefore, the CO₂RR of 2D c-MOFs on CNT/Au electrode needs to be checked whether they show the similar CO₂RR trend with that of 2D c-MOFs on CNT which they reported in the manuscript.

[1] Z. Weng, et al., Nat. Comm., 9 (2018)

[2] D.-H. Nam, et al., JACS, 140 (2018)

[3] S. Lin, et al., Science, 349 (2015)

Reviewer #2 (Remarks to the Author):

In this paper the authors reported bimetallic two-dimensional conjugated metal-organic framework (2D c-MOF) for the electrochemical conversion of CO₂ to syngas with controlled CO/H₂ ratios. In addition to the electrochemical measurements, in-situ IR characterization and DFT calculations were also performed to understand the reaction mechanisms. Overall this is a well written manuscript on an important research topic. The authors should address the following questions before the manuscript can be considered for publication:

1. The carboxyl intermediate (*COOH) is generally considered to be more important than the formate (HCOO*) intermediate over transition metal surfaces. However, it is less clear whether the formate intermediate would play an important role in CO₂RR in the current catalytic system involving Cu, Zn and MOF structures. The authors should consider DFT calculations to compare the activation barriers for the dissociation of *COOH and of HCOO* to confirm that the former is the key intermediate for CO₂RR.

2. The description of the near-edge regions of the Zn K-edge (Figure 2e) is unsatisfactory. The spectra from 5 samples are shown in Figure 2e, with several showing different line-shapes and different peak height of the 1s → 4p transition. Yet only one sentence was provided in the manuscript: "The Zn K-edge spectrum of PcCu-O8-Zn indicates a characteristic peak of 171 Zn(II) at 9665 eV (Fig. 2e), which is attributed to the 1s to 4p electronic transitions." The author need to explain more on the differences in the near-edge spectra.

3. Even though the authors stated that only gaseous H₂ and CO were detected, it would be useful for the authors to include a panel in Figure 3 showing the total Faradic efficiency (CO + H₂) at different

applied potentials. In addition, the name of the catalyst should also be mentioned in the caption of Figure 3d.

4. In the SEIRA results, vibrational features were not discussed for the carboxyl (*COOH) species that was identified as the key intermediate in the DFT calculations. The authors should offer reasons why this intermediate was not observed experimentally. Otherwise the SEIRA measurements and DFT calculations appeared to be unrelated.

Detailed Responses to the comments of the Reviewers

Reviewer #1

General Comments: *The authors report bimetallic two-dimensional conjugated metal-organic frameworks (2D c-MOF) applied for CO₂ reduction reaction (CO₂RR) electrocatalysts.*

They made 2D c-MOFs which spatially disperse single-metal-atom sites with different coordinate structures such as N-coordinated or O-coordinated sites in a single MOF. Specifically, their 2D c-MOF is composed of Cu-phthalocyanine as a ligand (CuN₄) of MOF and Zn-bis(dihydroxy) complex (ZnO₄) as a linkage of MOF. They sought to study the role of each individual atomic site as an active site for CO₂RR and hydrogen evolution reactions (HER) by matching trends in CO₂RR performance, DFT calculations, and operando surface-enhanced infrared-absorption spectroelectrochemistry (SEIRA). As the interest in single atom catalysts (SACs) is growing in electrocatalysis, this approach is of potential interest.

However, current manuscript has a serious problem because the exact nature of the material (especially 2D c-MOF) during CO₂RR has not been studied properly. This means that the DFT calculations and operando SEIRA (especially for intermediate detection) do not support the mechanistic proposal.

Also, the CO₂RR performance is not impressive.

Response: We appreciate Reviewer#1's encouragement and valuable comments. We agree that MOFs are of great potential in electrocatalysis. Particularly, as a new generation of MOFs, layered two-dimensional conjugated MOFs (2D c-MOFs), which possess higher density of exposed metal centers and improved electronic conductivity as well as the inherited features of traditional MOFs, hold great promise in high-performance electrocatalysis. In addition, the integration of molecular/atomic catalysts into the well-defined structures of the 2D c-MOFs affords a possibility for elucidating the intrinsic active sites and the reaction mechanism, benefiting for the design of high-performance electrocatalysts.

According to the constructive comments of Reviewer#1, operando X-ray absorption spectroscopy (XAS) of PcCu-O₈-Zn/CNT on carbon paper (CP) has been performed to gain profound insights into the nature of the catalytic sites during the CO₂RR process in our revised manuscript. *Ex-situ* Fourier-transform infrared-absorption (FT-IR) and Raman spectroscopy have been also utilized to investigate the stability of the PcCu-O₈-Zn/CNT. The combination of operando XAS, operando surface-enhanced infrared-absorption (SEIRA) spectroscopic studies, and *ex-situ* FT-IR, Raman and XRD measurements as well as contrast electrochemical testing and density functional theory (DFT) calculation demonstrates that the bimetallic MOF electrocatalyst (PcCu-O₈-Zn) is promising in enhancing CO₂RR: ZnO₄ units serve as the high catalytic active site for CO₂-to-CO conversion while CuN₄ complexes in the phthalocyanine macrocycles act as the synergetic component to promote the protonation process and hydrogen generation along with the CO₂RR.

The additional XAS data and contrast electrochemical testing will be discussed in the following response to Comment 2. The relevant revision has been added in the updated manuscript (Fig. 4 and Supplementary Fig. 20-22).

In terms of catalytic performance, we agree with the Reviewer#1 that PcCu-O₈-Zn/CNT is not the state-of-art high-performance example by taking into account of the reported metal, metal oxides or heteroatom-doped carbon catalysts. But our work demonstrates that the achieved sample shows superiority in CO₂RR to CO in comparison with the reported Zinc-based and MOF electrocatalysts (Supplementary Table 8), which highlights the potential in developing conductive 2D *c*-MOFs for energy-related technologies. We hope that our additional experiments and discussion will address your concerns on the catalytic origin here.

Comment 1: Authors defined TOF as ‘turn of frequency’ in both abstract (page 2) and introduction (page 4). TOF is ‘turnover frequency’, not ‘turn of frequency’.

Response: We thank Reviewer#1 for pointing this out and we feel sorry for such a mistake. The definition of TOF has been changed to “turnover frequency” in the revised manuscript.

Comment 2: Authors proceeded the DFT calculation (Figure 1, Figure 5) by assuming the structure of isolated Cu atom and isolated Zn atom in their 2D *c*-MOF. However, they didn't provide information about the real structure of 2D *c*-MOF during CO₂RR. We need to know whether the MOF is stable or not during CO₂RR. In the experience of most researchers, it is

not, i.e. the MOF changes dramatically in electrolyte and under reducing conditions, especially over extended operating conditions. Although they reported the SEM and XRD (Figure S18) after CO₂RR, this is not enough to know the real structure of the material.

Comment 2-1. Operando XAS experiment during CO₂RR is required for both Cu K edge and Zn K edge. Their XAS result (Fig. 2e, f, Fig. only shows the result of as-synthesized 2D c-MOF. Especially it has been reported in several papers that Cu ion (coordinated in N and O) is reduced to Cu (0) and Cu clusters form from organometallic complex and MOF during CO₂RR. [1,2]

Response: We agree with the Reviewer that operando XAS experiment is a strong tool to identify changes of the active sites during the catalytic process. In the revised manuscript, operando XAS experiments of PcCu-O₈-Zn/CNT were carried out to gain more insight into the valence states and coordination structures of the Cu and Zn centers, respectively, during the CO₂RR testing (Fig. R1). The results indicate that no significant change in both Cu and Zn K-edges, or no visible metallic Cu (0) and Zn (0) signals are observed in the X-ray absorption near-edge structure (XANES) profiles of PcCu-O₈-Zn/CNT during the catalytic process.

Fig. R1 | Operando XAS setup used in CO₂RR experiments. CE, RE, and WE stand for counter, reference and working electrodes, respectively. Operando XANES and EXAFS experiments were carried out at the BL14W1 beamline of the Shanghai Synchrotron Radiation Facility (SSRF).

Fig. R2 | Operando XAS measurement under electrolysis. a Zn K-edge XANES spectra of Zn foil, ZnO and PcCu-O₈-Zn samples. **b** Cu K-edge XANES spectra of Cu foil, CuO and PcCu-O₈-Zn samples.

As shown in Fig. R2a, the pre-edge signal of Zn(0) which usually appears at around 9660 eV (*J. Supercond. Nov. Magn.* 2010, 24, 659-663; *ACS Cent. Sci.* 2017 3 847-852), is not observed in the Zn K-edge XANES spectra of all PcCu-O₈-Zn/CNT samples. This excludes the possibility of the generation of metallic Zn in PcCu-O₈-Zn/CNT electrocatalyst during the CO₂RR process. Importantly, the main absorption peak at 9665 eV was not shifted in the Zn K-edge XANES spectra of PcCu-O₈-Zn/CNT as the applied potential was decreased to -0.4 and -0.7 V vs RHE and then increased back to the initial open circuit voltage (OCV) (Fig. R2a). The results reveal that the oxidation state of Zn(II) in PcCu-O₈-Zn/CNT is maintained throughout the catalytic process, which can be explained by the fact that the Zn(II) already has a full 3d electron shell (*ACS Cent. Sci.* 2017 3 847-852). In addition, the pre-edge peak at 8985 eV and the main absorption peak at 8998 eV in the Cu K-edge XANES spectra of PcCu-O₈-Zn/CNT (Fig. R2b) are not shifted with the decrease of the applied potential from OCV to -0.4 and -0.7 V vs RHE and then back to OCV, which indicates that the valence state of Cu(II) is also maintained during CO₂RR process (*Nat. Commun.* 2018 9, 415, *J. Am. Chem. Soc.* 2018, 140, 11378-11386). Notably, the missing pre-edge peak at 8980 eV (which is typical signal for Cu(0)) in Cu K-edge XANES spectra of all PcCu-O₈-Zn/CNT samples further confirms that no metallic Cu was generated at PcCu-O₈-Zn/CNT electrode under turnover condition. Therefore, the *operando* XAS results prove that the well-defined sites (ZnO₄ and CuN₄) act as the catalytic center during the CO₂RR process; while no metals or metal clusters formed via the reduction of high-valence metal centers. **(The relevant experiments and discussion have been added to**

our revised manuscript: Line 8-11 on Page11; Line 1-13 on Page 12, Fig. 4a-b and Supplementary Fig. 21)

Comment 2-2. In operando EXAFS data, the coordination number and atomic distance of coordinated atom around Cu and Zn need to be provided by EXAFS fitting. This result is important if they want to prove the N-coordinated, O-coordinated metal atomic site during CO₂RR.

Response: To monitor the local coordination environment changes, *in-situ* extended X-ray absorption fine structure (EXAFS) measurements were performed (Fig. R3). As the applied potential was decreased from initial (black curve) open circuit voltage (OCV) to -0.4 (red curve) and -0.7 V (blue curve) vs RHE, respectively, and then increased back (indigo curve), the peak at 1.55 Å assigned to Zn-O bond length in PcCu-O₈-Zn/CNT was not shifted (Fig. R3a). Meanwhile, the peak intensity presented a negligible decrease (black and indigo curves in Fig. R3a), which is possibly due to the interaction of the reaction intermediates and the ZnO₄ sites during the catalytic process, such as *H, *COOH, *CO and so on (*ACS Cent. Sci.* 2017, **3**, 847-852). Therefore, the above *in-situ* EXAFS results reveal no obvious change in Zn coordination number and bond length of Zn-O for PcCu-O₈-Zn/CNT under the electrolysis condition. Moreover, the characteristic signal of Zn-Zn bonding at 2.27 Å, was not observed in the EXAFS spectra of all the PcCu-O₈-Zn/CNT samples, which excluded the formation of metallic Zn or Zn cluster in PcCu-O₈-Zn/CNT catalyst throughout the CO₂RR process. For the CuN₄ complexes, no obvious shift of the Cu-N coordination peak at 1.54 Å was detected in the Cu K-edge EXAFS spectra of PcCu-O₈-Zn/CNT (Fig. R3b) upon performing the potential in one cycle. Additionally, the peak of Cu-Cu bonds at 2.23 Å was not found in all PcCu-O₈-Zn samples, which demonstrates that no heavy backscattering atoms (Cu) are bounding to Cu sites. Therefore, the N-coordinated and O-coordinated metal atomic sites remained stable during the CO₂RR process. (The relevant experiments and discussion have been added in our revised manuscript: Line 13-28 on Page12; Line 1 on Page 13, Fig. 4a-d and Supplementary Fig. 21.)

Fig. R3 | Operando XAS measurement under electrolysis. a Zn K-edge Fourier transform EXAFS spectra of Zn foil, ZnO and PcCu-O₈-Zn samples. **b** Cu K-edge Fourier transform EXAFS spectra of Cu foil, CuO and PcCu-O₈-Zn samples.

Comment 2-3. In XANES data, the magnified XANES spectra at the edge energy region need to be provide. In current XANES (Figure 2e), it is hard to distinguish the XANES positions according to samples.

Response: Thanks the reviewer for the valuable comment. We have added the magnified XANES spectra of PcCu-O₈-Zn in the revised manuscript. **(The relevant experiments and discussion have been added in our revised manuscript: Line 12-20 on Page 8, Fig. 2e)**

Generally, two characteristic signals are observed in the Zn XANES spectra including the pre-edge peak at around 9660 eV and the main absorption peak at 9660~9680 eV, which correspond to the electron transition from 1s to 3d (typically found for the transition metal Zn) and the 1s to 4p electronic transition, respectively (*J. Polym. Res.* 2016, **23**, 265; *Phys. Rev. Lett.* 2006 **97**, 037203). As shown in Fig R4, compared to the Zn foil, the pre-edge peak signal at 9660 eV is not found in the Zn K-edge spectrum of PcCu-O₈-Zn due to the full occupied 3d orbital of Zn²⁺, therefore excluding the existence of Zn(0) in PcCu-O₈-Zn. In addition, PcCu-O₈-Zn also shows a main peak at 9665 eV similar to the ZnCO₃ and ZnO (Fig. R5), which suggests the oxidation valence of Zn atom as +2 in PcCu-O₈-Zn.

Fig. R4 | Structure of PcCu-O₈-Zn. Zn K-edge XANES spectra for Zn foil, ZnO, ZnCO₃, PcZn(II) and PcCu-O₈-Zn. Inset: the enlarged figure of the XANES spectra between 9650 eV and 9700 eV.

Comment 2-4. FT-IR or Raman study during CO₂RR or after CO₂RR is required to prove the stability of MOF (especially organic materials).

Response: Thanks for the constructive suggestions. The stability analysis of the PcCu-O₈-Zn/CNT after CO₂RR by *ex-situ* FT-IR and Raman has been performed in our revision.

The stability of PcCu-O₈-Zn/CNT after long-term CO₂RR testing was gleaned from the absence of spectral changes in the Raman spectra before and after applying electrolysis. For example, the typical bands at 748, 1125 and 1456 cm⁻¹, corresponding to the macrocycle ring stretch, pyrrole breathing and metal associated with C-N-C vibration, respectively (Fig. R5a), were not changed in the Raman spectra of PcCu-O₈-Zn/CNT under turnover conditions. Additionally, comparison of the peak for C-O-Zn at 1272 cm⁻¹ in the FT-IR spectra (Fig. R5b) of PcCu-O₈-Zn/CNT before and after the long-term CO₂RR testing further demonstrated the retention of the integrity of the coordination networks and thus the stability of PcCu-O₈-Zn/CNT. **(The relevant experiments and discussion have been added to our revised manuscript: Line 1-14 on Page S28 in Supplementary information, Supplementary Figure 20c-d.)**

Fig. R5 | Stability studies of PcCu-O₈-Zn/CNT after CO₂RR testing in CO₂-saturated 0.1 M KHCO₃. a Raman spectra. b FT-IR spectra.

Comment 3: For the CO₂RR stability measurement (Fig. 3e), more CO FE data points are required between start and end point.

Response: Thanks for the valuable suggestion. More Faradaic efficiency data points of CO for PcCu-O₈-Zn/CNT have been added. Fig. R6 shows the stability of PcCu-O₈-Zn/CNT toward CO₂RR to CO after 10 h testing. **(Fig. 3e.)**

Fig. 6 | CO₂RR performance. Amperometry ($i \sim t$) stability and the according Faradaic efficiency for CO of PcCu-O₈-Zn/CNT at -0.7 V vs. RHE in CO₂-saturated 0.1 M KHCO₃.

Comment 4: When compare the CO₂RR performance with other MOFs related materials, current density, potential should be provided in the table (Table S8). Also, it seems that not all the results of MOFs based CO₂RR electrocatalyst are not included in the table. For example, a paper that reported the Co porphyrin-based COF for CO₂RR (90% CO FE, 24 h stability) has not been included in the table. [3]

Response: Thanks for the valuable suggestions. We have added the current density and potential of the related MOF and Zn based catalysts in the revised Supplementary Table 8. In terms of the current density, potential, FE for CO, PcCu-O₈-Zn /CNT shows superiority in electrocatalytic CO₂RR to CO toward syngas production than other reported MOF-based electrocatalysts.

Our previous **Table 8** in the Supplementary Information majorly provided a comparison of the CO₂RR performance of the thus-far reported MOFs and Zn based catalysts. The Co porphyrin-based covalent organic framework (COF) was not involved. According to the suggestion from Reviewer#1, we have added the COF-366 and COF-367 (*Science* 2015 349, 1208-1213) as a further contrast.

Table R1 Comparison of the CO₂RR performance of our 2D *c*-MOF with other related materials.

Materials	Electrolyte	Current Density (mA cm ⁻²) ^a	Potential (V vs RHE) ^b	FE for CO (%)	TOF s ⁻¹	Reference
PcCu-O ₈ -Zn /CNT	0.1 M KHCO ₃	8	-0.7	88	0.39	This work
Al ₂ (OH) ₂ TCPP-Co	0.5 M K ₂ CO ₃	2.3	-0.7	76	0.05	J. Am. Chem. Soc. 2015, 137, 14129-14135.
Fe_MOF-525	1 M TBAPF ₆ MeCN	5.9	-1.3 ^c	54	0.13	ACS Catal. 2015, 5, 6302–6309
COF-300-AR on Ag foil	0.1 M KHCO ₃	2.5	-0.85	80	/	Chem. 2018, 4, 1–14.
Fe/NG-750	0.1 M KHCO ₃	2.75	-0.6	80	/	Adv. Energy Mater. 2018, 1703487
ZIF-A-LD/CB	0.1 M KHCO ₃	3.2	-1	75	/	Angew.Chem.Int.Ed. 2019,58,4041 - 4045
PD-Zn/Ag	0.1 M KHCO ₃	-9.7	-1.2	74	/	Angew.Chem.Int.Ed. 2019,58,2256–2260
h -Zn	0.5 M KHCO ₃	-9.5	-0.95	85.4	/	Angew.Chem.Int.Ed. 2016,55,9297–9300
Ag@Al-PMOF	0.1 M KHCO ₃	-0.6	-1.1	55.8	0.014	Angew.Chem.Int.Ed. 2019,58,12632 -12639

COF-366-Co	0.5 M KHCO ₃	-5 mA mg ⁻¹	-0.67	90	0.69	Science 2015 349, 1208-1213.
COF-367-Co(10%)	0.5 M KHCO ₃	-4.5 mA mg ⁻¹	-0.67	70	2.6	

$$TOF = v_{CO} * flow\ rate * \frac{P_0}{RTn_{cat}}$$

Here, n_{cat} is the amount of the Zn of PcCu-O₈-Zn.

^a Current density is the highest partial current density for CO over the applied potentials.

^b The potential is which the catalyst shows the highest FE. ^c V vs. NHE

Comment 5: In this work, operando SEIRA analysis was done with 2D c-MOFs deposited onto a nanostructured Au surface. Although the authors mentioned 2D c-MOFs as a ‘closed film’, Au underlayer can affect the CO₂RR because Au is CO₂RR catalyst. 2D c-MOFs seem not fully continuous to cover all the surface area when I see the SEM images (Figure S6, S11, S12, S13). Therefore, the CO₂RR of 2D c-MOFs on CNT/Au electrode needs to be checked whether they show the similar CO₂RR trend with that of 2D c-MOFs on CNT which they reported in the manuscript.

[1] Z. Weng, et al., Nat. Comm., 9 (2018)

[2] D.-H. Nam, et al., JACS, 140 (2018)

[3] S. Lin, et al., Science, 349 (2015)

Response: We fully understand the concerns of the Reviewer#1, and we tried to avoid the influence from the Au substrate. When preparing the electrode, Nafion was added into the catalyst ink to fix the catalyst on the substrate. Herein, the 2D c-MOF catalysts with the Nafion can be evenly deposited as a closed film onto a nanostructured Au surface, which acted as IR signal amplifier. As for SEM measurement, the powder samples of 2D c-MOFs catalysts were directly utilized without any modification and additives. We apologize that the relevant SEM images have led to a misunderstanding that the 2D c-MOFs was not fully continuously covering the substrate.

To check if the Au substrate affects the CO₂RR performance of 2D c-MOF catalysts, we additionally prepared the nano Au on carbon paper (Au/CP) as the current collector and the performance of Au/CP and PcCu-O₈-Zn/CNT on Au/CP was evaluated. As shown in Fig. R7 Au/CP displays rather low CO₂RR performance with low Faradaic efficiency (FE) for CO (<16 %) and high FE toward H₂ (close to 90%) over the whole applied potentials. In contrast,

the PcCu-O₈-Zn/CNT on Au/CP exhibits much higher CO₂RR performance with Faradaic efficiency (FE) for CO of 89% at -0.7 V vs. RHE. The partial current density of CO for PcCu-O₈-Zn/CNT on Au/CP also presents a maximum value at -1.0 V vs. RHE, while H₂ generation (j_{H_2}) displays a steady rise over the increased overpotential because of the competitive reactivity between the CO₂RR and HER as well as the limitation of the transport of CO₂ to the catalytic sites. This trend of current density and FE_{CO} for PcCu-O₈-Zn/CNT on Au/CP is similar with PcCu-O₈-Zn/CNT on CP (Fig. R7 and R8), indicating the major contribution of PcCu-O₈-Zn/CNT to the CO₂RR performance of the whole electrode. Therefore, operando SEIRA analysis performed with 2D *c*-MOFs deposited onto a nanostructured Au surface indeed demonstrates the CO₂RR activity of 2D *c*-MOFs. (The relevant experiments and discussion have been added to our revised manuscript: Line 1-15 on Page S30, Line1-2 on Page S31 in Supplementary, and Supplementary Fig. 22. The literature listed by the reviewer#1 has also been cited as Ref. 11, 17, 40).

Fig. R7 | Comparison of electrochemical performance of Au/CP and PcCu-O₈-Zn/CNT on Au/CP in CO₂-saturated 0.1 M KHCO₃ at different potentials. a Partial current density and FE of CO for Au/CP; **b** Partial current density and FE of H₂ for Au/CP. **c** Partial current density and FE of CO for PcCu-O₈-Zn/CNT on Au/CP; **d** Partial current density and FE of H₂ generation for PcCu-O₈-Zn/CNT on Au/CP.

Fig. R8 | CO₂RR performance. a partial current and **b** Faradaic efficiency of CO for PcCu-O₈-Zn/CNT on CP at different potentials.

Reviewer 2:

General Comments: *In this paper the authors reported bimetallic two-dimensional conjugated metal-organic framework (2D c-MOF) for the electrochemical conversion of CO₂ to syngas with controlled CO/H₂ ratios. In addition to the electrochemical measurements, in-situ IR characterization and DFT calculations were also performed to understand the reaction mechanisms. Overall this is a well written manuscript on an important research topic. The authors should address the following questions before the manuscript can be considered for publication:*

Response: We appreciate t Reviewer#2 for the encouraging comment on our work and positive recommendation for publication after revision.

Comment 1: *The carboxyl intermediate (*COOH) is generally considered to be more important than the formate (HCOO*) intermediate over transition metal surfaces. However, it is less clear whether the formate intermediate would play an important role in CO₂RR in the current catalytic system involving Cu, Zn and MOF structures. The authors should consider DFT calculations to compare the activation barriers for the dissociation of *COOH and of HCOO* to confirm that the former is the key intermediate for CO₂RR.*

Response: We thank the reviewer#2 for pointing out this important issue. According to this comment, the activation barriers for *OOCH on PcM-O₈-M1 system was additionally calculated and related results/discussion have been provided in the revised manuscript (Supplementary Fig. 2, Line 8-11 on Page S2, Line 7-11 on Page S10 in Supplementary Information).

Generally, formate is formed with high selectivity only at a high overpotential on most materials (*J. Phys. Chem. Lett.* 2013 4, 388-392). Fig. R9 shows that the calculated barrier energy for the formation of the formate (*OOCH) intermediate is much higher than for the formation of the carboxyl intermediate (*COOH) on PcM-O₈-M1 catalyst with an energy difference of 3.834 eV on M1 site and 5.348 on M site. Therefore, the formation of formate was not favorable in PcM-O₈-M1 system.

Fig. R9 | Model structure of 2D *c*-MOFs with the intermediate (*COOH and *OOCH on M1 and M sites. The top view atomistic structure of the interaction of PcM-O₈-M1 carboxyl intermediate (*COOH, a: M1, c: M) and formate intermediate (*OOCH b: M1, d: M); The side view atomistic structure of the interaction of PcM-O₈-M1 carboxyl intermediate (*COOH, e: M1, g:M) and formate intermediate (*OOCH, f: M1, h: M).

Comment 2: The description of the near-edge regions of the Zn K-edge (Fig. 2e) is unsatisfactory. The spectra from 5 samples are shown in Fig. 2e, with several showing different line-shapes and different peak height of the 1s → 4p transition. Yet only one sentence was provided in the manuscript: "The Zn K-edge spectrum of PcCu-O₈-Zn indicates a characteristic peak of 171 Zn(II) at 9665 eV (Fig. 2e), which is attributed to the 1s to 4p electronic transitions." The author needs to explain more on the differences in the near-edge spectra.

Response: We feel sorry for such brief discussion about the near-edge regions of the Zn K-edge. More detailed explanation has been added to the revised manuscript. (Line 12-20 on Page8, Fig. 2e)

Generally, two characteristic signals are observed in the Zn XANES spectra including the pre-edge peak at around 9660 eV and the main absorption peak at 9660~9680 eV, which correspond to the electron transition from 1s to 3d (typically found for the transition metal Zn) and the 1s to 4p electronic transition, respectively (*J. Polym. Res.* 2016, **23**, 265; *Phys. Rev.*

Lett. 2006 97, 037203). As shown in Fig R4, compared to the Zn foil, the pre-edge peak signal at 9660 eV is not found in the Zn K-edge spectrum of PcCu-O₈-Zn due to the full occupied 3d orbital of Zn²⁺, therefore excluding the existence of Zn(0) in PcCu-O₈-Zn. In addition, PcCu-O₈-Zn also shows a main peak at 9665 eV similar to the ZnCO₃ and ZnO (Fig. R5), which suggests the oxidation valence of Zn atom as +2 in PcCu-O₈-Zn.

Comment 3: Even though the authors stated that only gaseous H₂ and CO were detected, it would be useful for the authors to include a panel in Fig. 3 showing the total Faradic efficiency (CO + H₂) at different applied potentials. In addition, the name of the catalyst should also be mentioned in the caption of Figure 3d.

Response: Thanks for the constructive suggestion. We have included the total Faradic efficiency of CO and H₂ in revised manuscript and Supplementary Information (Fig. 3c and Supplementary Fig. 18). And the names of the catalyst have been added in caption of Fig. 3d. The total FE (CO + H₂) is 99±2.2% (Fig. R10 and R11), further suggesting that only CO + H₂ were the products for the CO₂RR in aqueous electrolyte.

Fig. R10 | Electrochemical performance. Faradaic efficiency of CO and H₂ for PcCu-O₈-Zn/CNT, PcCu-O₈-Cu/CNT, PcZn-O₈-Zn/CNT and PcZn-O₈-Cu/CNT at -0.7 V vs RHE.

Fig. R11 | Electrochemical performance. Faradaic efficiency of H₂ and CO for PcCu-O₈-Zn/CNT **a**, PcCu-O₈-Cu/CNT **b**, PcZn-O₈-Zn/CNT **c** and PcZn-O₈-Cu/CNT **d** at different potentials.

Comment 4: In the SEIRA results, vibrational features were not discussed for the carboxyl (*COOH) species that was identified as the key intermediate in the DFT calculations. The authors should offer reasons why this intermediate was not observed experimentally. Otherwise the SEIRA measurements and DFT calculations appeared to be unrelated.

Response: It would have been very interesting to monitor the *COO⁻/H formation and particularly its conversion as a function of potential, etc. However, the detection of the *COO⁻/H using SEIRA spectroscopy (and in general surface-enhanced vibrational spectroscopy) is very challenging. Only few studies reported the detection of this or similar CO₂RR intermediates, and it is usually only possible under specifically chosen experimental conditions (*J. Am. Chem. Soc.* 2018,140, 4363-4371; *Proc. Natl. Acad. Sci. U. S. A.* 2018 115, E9261-E9270), which are not covered in our experimental design. In fact, in most SEIRA spectroscopic studies, the *COO⁻/H intermediate is not unambiguously identified/monitored,

even on bare Au/Cu surfaces where higher IR signal intensity is expected due to direct binding of the species to the IR signal-enhancing surfaces and higher density of surface active sites. Additionally, in our study, several more issues could prevent the observation of this intermediate. First, we worked in standard 0.1 M KHCO₃ electrolyte to be most comparable to the CO₂RR activity measurements and other systems in the literature. Both the HCO₃⁻ as well as the CO₃²⁻ anion show typically broad IR bands similar to the *COO⁻/H species. Due to the abundant concentration of the electrolyte (expected to be much higher than any transient concentration of the *COO⁻/H species), bicarbonate, carbonate and water migration to or away from the electrode surface induced by the applied potential will strongly impair a clear detection of the *COO⁻/H species. Second, the intensity of the *COO⁻/H species in the SEIRA spectra will depend on the ratio of rate constants for its formation and subsequent conversion, *i.e.* protonation and water elimination, respectively. It is expected that the *COO⁻/H formation rate will be slow (rate-determining-step) as also confirmed by our DFT results. The subsequent protonation proceeds downhill in energy and is therefore expected to be (much) faster. In result, the transient concentration of the *COO⁻/H will likely be too low to be observed.

REVIEWERS' COMMENTS:

Reviewer #1 (Remarks to the Author):

The authors have acted on the request for in-situ/operando XAS for Cu K-edge and Zn K-edge studies. The tracking of oxidation states and atomic structure (atomic distance and coordination number) during CO₂RR indicates the stability of their 2D c-MOF. The coordinative structures around Cu and Zr do not change at -0.4 V nor at -0.7 V. I agree that ZnO₄ and CuN₄ can be active sites for the CO₂RR of 2D c-MOF. The request for FT-IR and Raman study was acted upon and added to the study the stability of organic molecules during CO₂RR was found, consistent with maintaining the MOF structure. I recommend the publication of this paper in Nature Communications.

Reviewer #2 (Remarks to the Author):

The authors have satisfactorily addressed all my questions. I recommend acceptance of the revised manuscript.